# The use of carbogen for interruption of febrile seizures - the randomized controlled CARDIF trial

Claudia Weiß[1,2], Alice Hucko[1,2], Stephanie Müller-Ohlraun[1], Stefanie Märschenz[1], Peter Martus[3], Dietmar Schmitz[4,5], Markus Schuelke[1,2,5]*

1 NeuroCure Clinical Research Center (NCRC), Charité - Universitätsmedizin Berlin, corporate member of Freie Universität Berlin and Humboldt-Universität zu Berlin, Berlin, Germany, 2 Department of Neuropediatrics, Children's Hospital, Charité - Universitätsmedizin Berlin, corporate member of Freie Universität Berlin and Humboldt-Universität zu Berlin, Berlin, Germany, 3 Institute for Clinical Epidemiology and Applied Biostatistics, Eberhard-Karls-Universität, Tübingen, Germany, 4 Neuroscience Research Center (NWFZ), Charité - Universitätsmedizin Berlin, corporate member of Freie Universität Berlin and Humboldt-Universität zu Berlin, Berlin, Germany, 5 NeuroCure Cluster of Excellence, Charité - Universitätsmedizin Berlin, corporate member of Freie Universität Berlin and Humboldt-Universität zu Berlin, Berlin, Germany

* markus.schuelke@charite.de

## Abstract

Febrile seizures are the most common seizure disorders in children. Fever-induced hyperventilation and subsequent hypocapnia may precipitate febrile seizures. In preclinical studies and in individual children, increasing $CO_2$ partial pressure has shown potential to terminate febrile seizures. We hypothesized that the use of carbogen (5% $CO_2$ plus 95% $O_2$) in the home environment would be an effective and safe treatment for recurrent febrile seizures. The CARDIF (CARbon DIoxide against Febrile seizures) trial is a randomized, monocentric, prospective, double-blind, placebo-controlled, crossover study to determine whether short-term inhalation of carbogen in the home environment can stop febrile seizures. 100% oxygen was used as a placebo control. We included children aged 0.5 to 5.0 years who presented after a first febrile seizure in the absence of severe organ or neurological disease, pathological EEG changes, or a history afebrile seizures. We noted parent-reported seizure duration, benzodiazepine use, and any serious adverse events. We enrolled n = 92 patients. In n = 20 children, at least one recurrent febrile seizure was treated with either *carbogen* or *oxygen*. Six of these children received both *carbogen* and *oxygen* in a planned crossover design. The febrile seizure was terminated in 5/15 episodes on *carbogen* and in 8/11 episodes on *oxygen* (Fischer's exact test; p = 0.11). Children with ≥2 recurrent febrile seizures completed the crossover arm. In these children, febrile seizures stopped during *carbogen* administration in 3/6 cases and during *oxygen* administration in 5/6 cases. In conclusion, carbogen did not interrupt acute febrile seizures more often than *placebo*. Home caregivers had difficulty determining when a seizure had stopped.

**Data availability statement:** Anonymized raw data from this study have been deposited in the Zenodo repository and are accessible at https://zenodo.org/records/15728814. All personal identifiers were removed in accordance with EU GDPR requirements for anonymization. No legal or ethical restrictions apply to the use or redistribution of these data.

**Funding:** The CARDIF trial was funded by the German Research Foundation (DFG, Deutsche Forschungsgemeinschaft) through the German Excellence Strategy (EXC-2049-390688087) via the NeuroCure Consortium at Charité - Universitätsmedizin Berlin to DS and MS and by the Collaborative Research Center SFB 665 TP C4 to DS and MS. The funders had no role in study design, data collection and analysis, decision to publish or preparation of the manuscript.

**Competing interests:** The authors have reported that they have no potential conflicts of interest to disclose.

**Abbreviations:** AE, adverse event; ASM, anti-seizure medication; CARDIF, CArbon, DIoxide against Febrile seizures; $CO_2$, carbon dioxide; CRF, case report form; EEG, electroencephalography; DFG, German Research Foundation (Deutsche Forschungsgemeinschaft); FS, febrile seizure; GEFS+, generalized epilepsy with febrile seizures +; NCRC, NeuroCure Clinical Research Center; $pCO_2$, partial pressure of carbon dioxide; $SaO_2$, oxygen saturation of arterial blood; SAE, serious adverse events.

## Trial registration

ClinicalTrials.gov NCT01370044

## Introduction

Two to nine percent of all children will have at least one febrile seizure in their lifetime, depending on ethnicity [1–3]. Of these children, 30–35% will have recurrent febrile seizures [2,4] and 18% will have particularly prolonged febrile seizures with a mean duration of 39.8 minutes [5], creating a very threatening situation for parents. Febrile *status epilepticus* with seizures lasting more than 30 minutes is associated with an increased risk of developmental delay and damage to the hippocampal formation [5,6]. The standard therapy for interrupting febrile seizures is rectal diazepam or buccal midazolam, administered after a 3–5 minute waiting period [7,8]. These benzodiazepines have a sedative effect and cause drowsiness. In addition, benzodiazepines have caused respiratory depression in 9% of patients [9] and were ineffective in 19% of patients who progressed to febrile *status epilepticus* in the FEBSTAT study [10]. The authors of the FEBSTAT study conclude that shortening the time from seizure onset to initiation of anti-seizure medication (ASM) was significantly associated with shorter seizure duration [10]. Therefore, an alternative therapy based on the pathophysiology of febrile seizures that is independent of ASM use would be preferable.

Ambient air contains 0.04% carbon dioxide ($CO_2$). In an animal study, Schuchmann *et al.* demonstrated that an increase of the inhaled air $CO_2$ content to 5% immediately terminated febrile seizures in a rat model of febrile seizures [11]. Similarly, inhalation of 10% $CO_2$ had a rapid anticonvulsant effect against hyperthermia-induced seizures in rats with a mutation in the sodium voltage-gated channel alpha subunit 1 (*Scn1a*) [12]. Zions *et al.* studied naked mole rats with variants in their *Slc12a5* gene, encoding for an integral membrane potassium plus chloride cotransporter. In these animals, hyperthermia induced seizures could also be blocked in the presence of high ambient $CO_2$ concentrations [13]. Mutations in the human ortholog (*KCC2*) of the *Slc12a5* gene mutations have been reported to cause febrile seizures and epilepsy in humans [14,15].

It has been hypothesized that fever stimulates the respiratory rate disproportionately in younger children as compared to older individuals [16], which may explain the characteristic time window, when febrile seizures occur between 0.5 and 5.0 years of age. In some individuals, this hyperventilation and subsequent respiratory alkalosis may precipitate seizures. In fact, Schuchmann *et al.* described that children admitted to the emergency department with febrile seizures were more likely to have alkalosis than children admitted with gastrointestinal infections [17]. Kilicaslan *et al.* confirmed in a prospective case-control study that the $pCO_2$ was significantly lower in children with febrile seizures than in children with febrile illness without seizures. Furthermore, $pCO_2$ levels were even lower in children with complex febrile seizures than in children with simplex febrile seizures [18]. Anecdotal case reports in the hospital setting described rapid seizure termination after rebreathing [19] or inhalation of 5% carbon dioxide [20].

These encouraging results prompted us to test this therapy, which appears to be rapidly effective in animal studies and has no reported adverse effects in humans, in a formal controlled clinical trial in children with febrile seizures. However, rebreathing through a plastic bag is too dangerous for home use due to the risk of uncontrolled increases in blood $pCO_2$ into the danger zone and the potential of hypoxia. As a safe alternative for home use, where pulse oxymeters are usually not available, we decided to use carbogen gas, a mixture of 5% carbon dioxide and 95% oxygen. The high oxygen content ensures that no asphyxia can occur, and the 5% carbon dioxide should induce the desired acidification of the blood pH and subsequently suppress the epileptic discharges in the brain. For this end, we developed a low-pressure can that delivers 6 liters of carbogen through an attached loose-fitting mask for home use. The device description and achievable end-tidal $pCO_2$ levels have been published previously [21].

## Methods

### Study design

**CA**rbon **DI**oxide against **F**ebrile seizures (CARDIF) was a randomized, double-blind, crossover, placebo-controlled, phase II/III trial to determine whether 6 liters of carbogen gas (5% carbon dioxide + 95% oxygen) delivered from a low-pressure can would suppress acute febrile seizures in an outpatient home setting (Fig 1). CARDIF was an investigator-initiated trial that was conducted at the NeuroCure Clinical Research Center (NCRC) of the Charité – Universitätsmedizin Berlin between 2012 and 2015, with a planned duration of 24 months for each study participant. The study was approved by the Ethics Committee of the Senate of Berlin (11/0209 – ZS EK 15) and by the German Federal Institute for Drugs and Medical Devices (BfArM, 61-3910-4037762), which is the competent authority for approval of studies under the German Medicines Law (Arzneimittelgesetz) and the Medical Devices Law (Medizinproduktegesetz). The trial has been pre-registered on ClinicalTrails.gov (NCT01370044) and in the EU Clinical Trials Register (EudraCT 2011-01403-12). The study design and the study protocol have been published previously [21] and are added as **Supplementary material** (S1 File). The study was conducted according to the guidelines of Consolidated Standards of Reporting Trials (CONSORT) and good clinical practice [22]. It was monitored every three months by an independent monitor.

### Hypothesis

Six liters of carbogen gas (5% carbon dioxide + 95% oxygen) administered over a period of 3 minutes through a low-pressure can with an attached loose-fitting face mask is safe and more effective than *placebo* (100% oxygen) to interrupt an acute ongoing febrile seizure within 3 minutes.

### Inclusion and exclusion criteria

We recruited children aged 0.5 to 5.0 years of both sexes with a history of at least one simple or complex febrile seizure as defined by the American Academy of Pediatrics [23]. Exclusion criteria were **(i)** other serious organ disease, **(ii)** neurological disease or developmental delay, **(iii)** pulmonary disease (e.g., asthma), **(iv)** a pathological EEG in the interval and cerebral seizures without fever, and **(v)** intolerance to carbogen (e.g., anxiety disorder).

### Randomization and allocation concealment

Because of the psychological and procedural problems with obtaining written informed consent from parents/guardians during the acute phase of a seizure, we had to recruit study participants during seizure-free intervals.

When children with a febrile seizure were admitted to the Emergency Department of the Charité – Universitätsmedizin Berlin, parents/guardians were informed about the study by the attending physician and through the use of information leaflets. Formal screening, recruitment, and informed consent were performed by a study physician. To recruit children with a higher likelihood of having a febrile seizure during the 24-month observation period, we randomized only children who had already had one or more previous febrile seizures, which increased the likelihood of subsequent seizures by 30–44% [24].

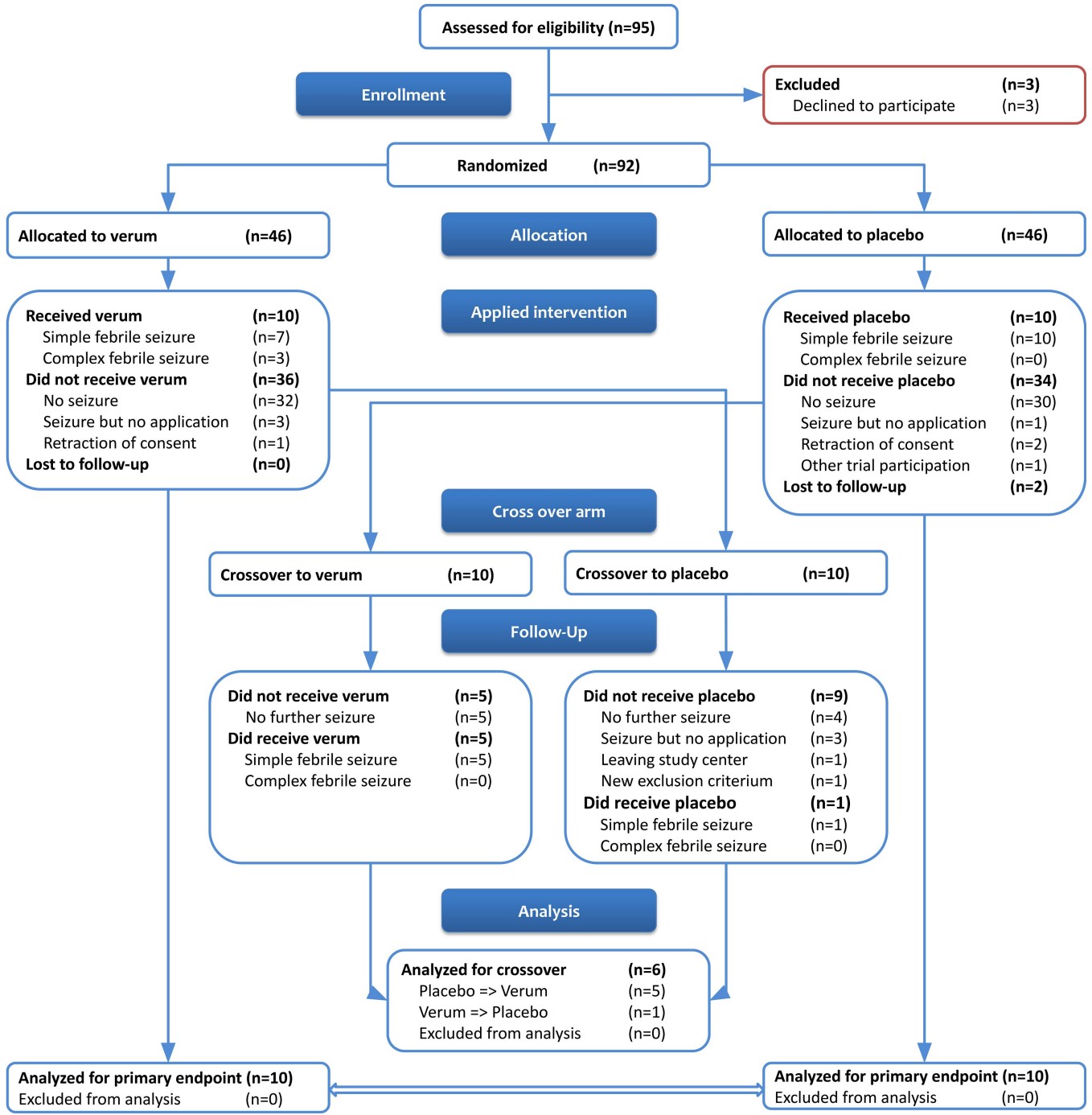

**Fig 1. CONSORT flow chart of the CARDIF study.**

Based on the "*modified intention-to-treat population*" principle [25], we had to randomize three times as many individuals (n = 288) as planned to enter the analysis (n = 20 seizure recurrences for the first study phase/interim analysis, n = 52 for the second study phase/final analysis). Patients were randomized to the "*first carbogen (verum)*" and "*first oxygen*

*(placebo)"* groups. After the first seizure, crossover occurred, e.g., patients who received *placebo* for the first seizure recurrence received *verum* for the second recurrence and *vice versa*.

### Study medication & appliance

The CE-certified low-pressure aluminum can was manufactured by MTI IndustrieGase AG (Neu-Ulm, Germany) and filled with 6 liters of medical carbogen (5% $CO_2$ + 95% oxygen; *verum*) or 6 liters of medical oxygen (*placebo*), fitted with a plastic face mask and labeled with serial numbers (Fig 2). After pressing the lever, the complete contents of the can emptied within 3 minutes.

### Blinding

The cans were labeled by the manufacturer. Each study participant was assigned two cans, one *carbogen* and one *oxygen*, with neither the patients, their caregivers nor the study team knowing which came first. For the third open-label continuation arm of the study, cans of carbogen (*verum*) were manufactured and labeled accordingly. Unblinding envelopes were kept at the study site for use if deemed necessary. The study personnel who enrolled and assigned participants to the intervention and outcome assessors were blinded as well.

### Intervention

Caregivers were advised to carry the can with them wherever they went with their child and were thoroughly trained in its use by the study staff. All caregivers received a graphic instruction about how to use the can (S2 File, Fig 2). In the event of a febrile seizure, caregivers were instructed to use the can immediately at seizure onset. Administration involved first attaching the face mask to the top of the bottle and placing the mask loosely over the child's mouth and

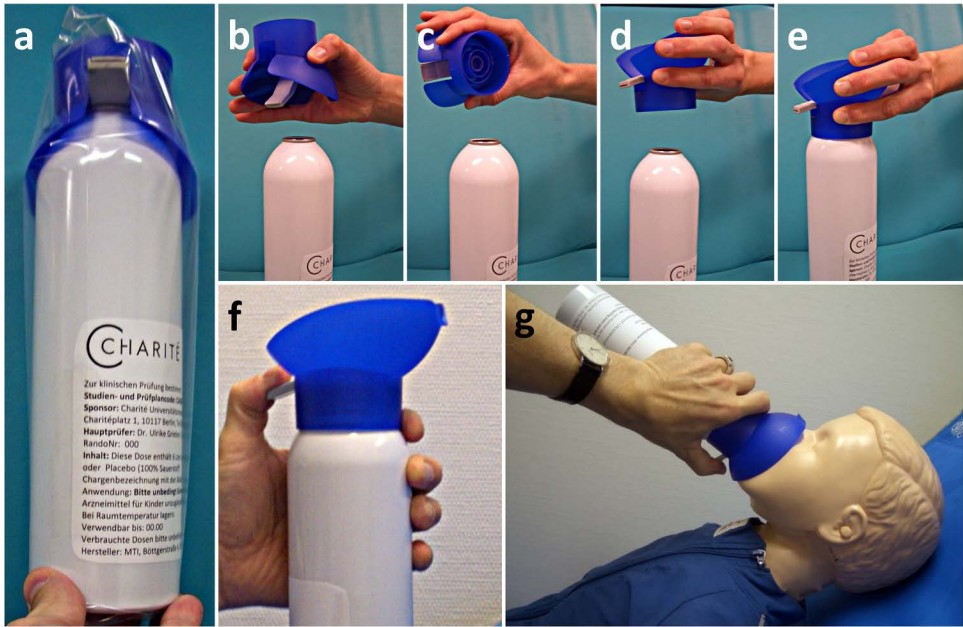

**Fig 2. Application of study medication.** (a) Originally packaged study medication with a blue breathing mask. For activation of the can, tear open and remove the plastic packaging, **(b)** lift off the mask, **(c-d)** turn the mask 180 degrees over, **(e)** place the mask firmly on the bottle by snapping it into place, **(f)** grasp the bottle as shown. When the gray lever is pressed, the gas begins to flow into the mask with a hissing sound, **(g)** hold the mask close to the child's face, but do not press it against the face. The gray lever on the mask must be pressed for the entire time (3 minutes).

nose. They were then instructed to press a lever on the side of the can to release the study medication. The escaping gas produced a hissing sound that stopped after 3 minutes, when the bottle had emptied itself. If the seizure did not stop during this time, caregivers were advised to use standard therapy and administer rectal diazepam. The ethical requirements imply that no standard treatment must be withheld from the study participants. This would imply the administration of benzodiazepines had the seizure not stopped within the first 3 minutes [26]. We therefore had an observational window of 3 minutes to observe the effect of the study medication, before the standard medication (benzodiazepines) had to be administered.

The necessity to provide standard drug treatment (e.g., diazepam) was considered a "failure" of the study medication. If the seizure stopped within the 3 minute time window of carbogen application, it was considered a "success" of the study medication. If the onset of a seizure was not observed and the duration was therefore unclear, this seizure was not included in the analysis. Caregivers were asked to report immediately to the study center each time a febrile seizure occurred to record the body temperature and whether seizure control was achieved. Thereafter, the caregivers received the crossover study medication.

Application of antipyretics (ibuprofen or acetaminophen) was recommended for all children with a temperature above 38.5°C as standard of care.

### Primary outcome measures

**Seizure-cessation efficacy endpoint**: The primary binary outcome measure was seizure cessation [yes|no] within the three minutes of the study medication application. The need to apply rectal diazepam at 3 minutes after the start of the seizure was recorded as a failure [no].

### Secondary outcome measures

**Safety**: All serious adverse events (SAEs) and all study medication-related adverse events (AEs) occurring during the trial were recorded: [yes|no] and additionally as free text [string].

**Feasibility**: Parents were asked about the practicability of handling and using the can and any problems encountered during its use: [yes|no] and additionally as free text [string].

### Power analysis

Because previously published clinical data on the use of carbogen to terminate cerebral seizures were rare, scattered, and not systematically recorded, we were unable to derive an appropriate effect size that we expected to achieve during the trial. Therefore, we chose the two-phase adaptive design of Bauer & Köhne [27] with $\alpha = 0.025$, $c_\alpha = 0.00380$ and $\alpha_0 = 0.5$ (one-sided), which included an adaptive interim analysis to allow for an upward modification of the sample size in a second phase of the trial without compromising the overall significance level and to ensure the overall power of the trial. In this manuscript, we present data from the interim analysis. The primary outcome of the trial "*interruption of the seizure within 3 minutes (success)*" *versus* "*application of diazepam (failure)*" was to be evaluated as a binary endpoint using Fisher's exact test. With an assumed difference of 75% *versus* 25% in 2*26 patients, the power for the two-sided Fisher's exact test is 86% (worst case $p_1 = c_\alpha/\alpha_0$). We were aware that an unequal distribution of seizure events between treatment arms, which might occur in studies with a low number of patients/ treatment incidents might impact the power analysis and increase the $\alpha$ (type I) error.

### Data collection and management

Data on the child's health status at randomization and during the study visits were collected on paper CRFs and transferred to an electronic database programmed in Filemaker v7.0. AEs and SAEs were also recorded in the same database.

## Statistical analysis

**Statistical analysis of endpoints.** Fischer's exact test was used to analyze the interruption of febrile seizures (primary outcome) by the study medication, the number of past febrile seizures, and the sex of study participants. As we only had n = 6 patients in the crossing over arm, statistical analysis was not feasible and the data are only presented as median and interquartile range. P-values for age at recruitment, age at febrile seizure, body temperature during febrile seizure, and seizure duration were calculated using the non-parametric two-tailed Man-Whitney U-test.

## Results

### Patient enrollment, study completion, and analysis

Between July 16, 2012, and April 22, 2015, n = 95 children with at least one febrile seizure were screened for eligibility at a single study center at Charité – Universitätsmedizin Berlin (Fig 2). N = 3 children were excluded because their caregivers did not consent to participation in the trial. Baseline characteristics of the n = 92 recruited patients are shown in Table 1. Participation in the study was terminated prematurely in n = 3 patients prior to study drug administration due to withdrawal of consent by caregivers, and in n = 1 patient due to participation in another study. N = 1 patient withdrew from the study due to relocation after he had already received the *verum* dose, and n = 1 patient had to be excluded due to meningitis, having already received at first the *verum* and then the *placebo*. N = 2 patients were lost to follow-up (Table 2). The trial was stopped for futility prematurely on June 30, 2015, due to the interim analysis that was negative for efficacy.

N = 23 children had at least one recurrent febrile seizure, ten out of whom had at least two recurrent febrile seizures (five children with three and one child with five recurrent seizures) giving a total of n = 38 febrile seizure events during the study. The study medication was not used during eight febrile seizure events: **(i)** in n = 6 cases, the study medication was not at hand because the child was in day care or the family was out shopping, **(ii)** in n = 2 cases, caregivers encountered problems in handling the study medication, which are detailed below.

Ultimately, the study medication was used in n = 30 febrile seizures in n = 20 children. In n = 6 children, more than one febrile seizure was treated with the study medication, allowing a crossover for the second application. Subsequently, two of these children received carbogen as *open verum* for a total of 4 febrile seizures (one child had one application of *open verum*, the other child had three) (Fig 1).

### Efficacy on seizure cessation

We had planned to analyze "*interruption of seizure within 3 minutes (success)*" versus "*use of diazepam (failure)*". However, in n = 2 seizures the caregivers applied diazepam after the end of a seizure, while in n = 3 other seizure events no diazepam was administered despite a seizure lasting more than 3 minutes. Therefore, the children who suffered a prolonged febrile convulsion were not entirely identical to those who received diazepam. We therefore decided to consider only the cessation of febrile seizures during the application of the study medication as the primary endpoint.

N = 15 febrile seizures were treated with carbogen (*verum*) during the trial period. 5/15 febrile seizures stopped during the 3 minutes application of carbogen, while 10/15 continued. Diazepam was used in n = 7 seizures lasting more than 3 minutes and in n = 2 seizures, even though the seizure had stopped within 3 minutes. In n = 2 seizures diazepam was not used despite a seizure duration of more than 3 minutes because the seizure stopped 1–2 minutes after the application of carbogen. In n = 11 febrile seizures, oxygen (*placebo*) was applied. In 8/11 cases, the febrile seizure stopped during the 3 minutes of *placebo* application (p = 0.11). Diazepam was administered in 2/11 cases with a seizure lasting more than 3 minutes, and in 1/11 cases the seizure stopped after 6 minutes without further intervention. In the *verum* arm n = 12 recurrent febrile seizures were simple and n = 3 were complex, whereas all recurrent febrile seizures in the *placebo* arm were simple. In n = 2 children, more than 2 recurrent febrile seizures occurred and were treated with *open verum* according to the protocol (n = 4 seizure occurrences in the *Verum open* arm). Of these, 2/4 seizures stopped during the application of

**Table 1. Baseline characteristics of recruited patients and characteristics of treated patients and febrile seizures.**

| | *Verum* (carbogen) double-blinded arm | *Placebo* (oxygen) double-blinded arm | *Verum* open arm | *p-value* |
|---|---|---|---|---|
| **Recruited patients, n** | 46 | 46 | | |
| one FS in history, *n* [%] | 32 [69.6] | 30 [65.2] | | 0.82* |
| more than one FS in history, *n* [%] | 14 [30.4] | 16 [34.8] | | |
| Age at recruitment, median [months] | 22 [IQR 11] | 20 [IQR 14] | | 0.63** |
| Male sex, *n* [%] | 20 [43.5] | 26 [56.5] | | 0.30* |
| Female sex, n [%] | 26 [56.5] | 20 [43.5] | | |
| **Patients with febrile seizure during study period [all uses of study medication]** | | | | |
| **Received study medication, n** | **15** | **11** | **4** | |
| Male sex, *n* [%] | 6 | 6 | 1 | 0.69* |
| Female sex, n [%] | 9 | 5 | 1*** | |
| Body temperature at time of FS [°C] | 39.5 [IQR 1.0] | 39.3 [IQR 1.3] | | 0.31** |
| Age at febrile seizure, median [months] | 25 [IQR 7] | 24 [IQR 9] | | 0.82** |
| Seizure duration [minutes] | 5 [IQR 2] | 3 [IQR 3] | | 0.11** |
| Seizure terminated [%] | 5 [33.3] | 8 [72.7] | 2 [50.0] | 0.11* |
| Seizure not terminated [%] | 10 [66.3] | 3 [27.3] | 2 [50.0] | |
| **Patients with ≥2 febrile seizures during study period [crossover]** | | | | |
| **Received study medication, n** | 6 | 6 | | |
| Male sex, n [%] | 3 [50.0] | 3 [50.0] | | |
| Female sex, n [%] | 3 [50.0] | 3 [50.0] | | |
| Body temperature at time of FS [°C] | median = 39.1 [IQR 1.0] | median = 38.9 [IQR 0.5] | | |
| Age at febrile seizure, median [months] | median = 23.0 [IQR 7] | median = 21.5 [IQR 7] | | |
| Seizure duration [minutes] | median = 3.5 [IQR 3.5] | median = 4.0 [IQR 2.5] | | |
| Seizure terminated, n [%] | 3 [50.0] | 5 [83.3] | | |
| Seizure not terminated, n [%] | 3 [50.0] | 1 [16.4] | | |

*Fischer's exact test; **two-tailed Man-Whitney U-test; IQR interquartile range; *** one girl received three times *open verum*

carbogen and diazepam was never required. Patients with at least two febrile seizure recurrences received sequential application of *verum* and *placebo* or *vice versa* in a crossover design and were of particular interest. This was the case in five children who received *placebo* at the first recurrence and *verum* at the second and in one child who received first *verum* and then *placebo* (Fig 1, Table 1). Of these six children who received both *verum* and *placebo*, febrile seizures stopped during *verum* administration in 3/6 cases and during *placebo* administration in 5/6 cases (p = 0.22). Body temperature, age at recurrent febrile seizure, and seizure duration were similar in the *verum* and *placebo* groups (Table 1).

## Safety analyses

N = 6 SAEs occurred in n = 5 patients. In all cases, the SAE was due to hospitalization after a complicated febrile seizure. One child was diagnosed with enterovirus meningitis approximately five months after application of the last dose of the study medication. Because of frequent febrile seizures, generalized epilepsy with febrile seizures + (GEFS+)

was suspected, and anti-seizure medication was started. This child was excluded from further participation in the study (Table 2).

### Practicability of the study medication

In one case, the can fell on the floor and the plastic mask broke, and in another case, no flow sound was heard, and the caregivers applied diazepam instead. In all other cases, the caregivers reported that the handling of the study medication was successful without any problems (Table 2).

## Discussion

The aim of the CARDIF study reported here was to interrupt febrile seizures by treating their pathophysiologic cause. The hypothesis was that fever-induced hyperventilation and subsequent respiratory alkalosis cause febrile seizures, and that carbon dioxide inhalation could reverse these effects, thereby interrupting the febrile seizure [11,17,19]. Because febrile seizures occur primarily at home, we developed a low-pressure can containing carbogen to safely deliver carbon dioxide in the home care setting.

However, the primary outcome measure was not met. Instead, at the time of the interim analysis, we found no significant differences in the rate of seizure interruption or seizure duration between the use of oxygen (*placebo*) and carbogen (*verum*). Therefore, the study was stopped prematurely for futility. One possible reason for seeing no differences between *carbogen* and *oxygen* treatment could be that the concentration of 5% $CO_2$ may have been too low for home use. This concentration had been chosen because inhalation of 6 liters of carbogen over 3 minutes at a respiratory rate of 40/min had increased end-tidal $pCO_2$ by 7 mmHg in our preliminary studies under ideal laboratory conditions [21]. However, in the stressful situation of a convulsing child at home, caregivers may not have been able to hold the face mask close enough to the face, resulting in less carbogen reaching the child's airway. Furthermore, the difference in complex versus simple febrile seizures in both study arms may have influenced the outcome as complex febrile seizures only occurred in the *verum* arm.

**Table 2. Severe Adverse Events [SAEs], study-related Adverse Events [AEs] and drop-outs.**

| | *Verum* (carbogen) applied, double-blinded arm | *Placebo* (oxygen) applied, double-blinded arm | *Verum* open arm | No study medication applied |
|---|---|---|---|---|
| **Serious adverse events (SAE), n** | 15 | 11 | 4 | 8 |
| Hospital admission after complicated febrile seizure, n [%] | 2 [13.3] | 1 [9.1] | 1 [25] | 1 [12.5] |
| Hospital admission after complicated febrile seizure and enterovirus meningitis, n [%] | 1 [6.7] 9 months before meningitis* | 1 [9.1] 5 months before meningitis* | | |
| **Study-related adverse events** | | | | |
| When mask was applied, can fell to the floor and mask broke, n | | 1 | | |
| No hissing sound was heard | | 1 | | |
| **Drop-outs** | | | | |
| Retraction of consent, n | | | | 3 |
| Relocation, n | | | | 1 |
| Participation at another trial, n | | | | 1 |
| New exclusion criterium (enterovirus meningitis*), n | | 1 | | |
| Lost to follow-up, n | | | | 2 |

*One patient suffering from recurrent febrile seizures first received *verum*, 4 months later *placebo* and another 5 months later suffered from an enterovirus meningitis with a complicated febrile seizure. This event was not considered to be study-related.

The use of carbogen is safe. There were no AEs or SAEs associated with the study medication. However, two families reported problems with the use of the investigational medicinal product: in one case, the caregivers found it difficult to attach the face mask correctly due to nasal discharge or saliva, and in a second case, it was difficult to hear the hissing sound of the gas flowing from the can. In addition, the study medication was not always at hand when a febrile seizure actually occurred, for example, when children were in day care, out shopping, or in the care of people who had not been trained to use the study medication. Because febrile seizures often occur during the initial rise in body temperature, it was not always possible for caregivers to recognize a febrile infection in time to carry the study medication along. Rebreathing into a bag would be an alternative way to increase $pCO_2$, but there were safety concerns about using such devices in a home care setting without the availability of $pO_2$ and $pCO_2$ monitoring to prevent hypoxia and hypercapnia. Diazepam rectal tubes, on the other hand, are smaller than the carbogen canister and guardians may find it easier to carry and administer. After our study had been concluded, the buccal application of midazolam in the home environment has become the standard treatment for febrile seizures as it is more effective than rectal diazepam [8] and also easier to apply. Although benzodiazepines may cause central nervous system adverse drug reactions such as drowsiness and mild ataxia, these may also be caused by the febrile infection and/or seizure itself [28].

As patient-centered approaches are a stated goal for the approval of new drugs by the European Medicines Agency (EMA) and the Food and Drug Administration (FDA), patient-reported outcome measures (PROMs) are becoming increasingly important [29]. In this study, we considered caregiver assessment of seizure duration as the primary endpoint for the efficacy of carbogen. However, it was very difficult for caregivers (usually parents) to detect the end of a febrile seizure in a stressful situation at home. This may explain why we could not reproduce the beneficial effect of $CO_2$ inhalation reported by Tolner *et al.* and Schuchmann *et al.* in a small pilot study conducted in a hospital setting under continuous electroencephalographic monitoring [19,20] and why the results of this study differ dramatically from the results seen in animal studies [11–13,16].

We believe that the main limitations of the present study are the difficulties for caregivers to administer the carbogen correctly in a convulsing child with large amplitude generalized tonic-clonic seizures and copious nasopharyngeal secretions and to accurately determine the duration of a seizure. Detecting the end of a seizure without an EEG recording may be difficult even for medically trained staff. Some children are tired, drowsy, and may fall asleep at the end of a febrile seizure, making it difficult to clearly distinguish an ongoing seizure from the immediate postictal period. In a situation that is perceived to be dangerous for the child, it is even more difficult for caregivers to clearly identify the end of a seizure. Ideally, outpatient EEG should be performed on all study participants. Since this is impractical over a long study period, seizure detection devices that record movement as well as heart rate could help clarify the end of a seizure. In recent years, portable devices ("wearables") have become available [30,31] that may help to detect the end of hypermotor seizures, whereas it would remain difficult to detect the end of tonic seizures.

The low-pressure can has been filled with carbogen or pure oxygen so that 6 liters will flow in 3 minutes, during which time a hissing sound should be heard. However, if the handle is not fully pressed down, the gas may flow more slowly, and the duration of the hissing sound may be prolonged or too low. In this case, the duration of the seizure may be slightly longer than indicated. On the other hand, the hissing sound may not be heard well in a noisy environment, so the end of the 3 minutes may not be easily recognized. In the future, the flowing sound could be augmented by a whistle and/or an attached electronic timer could be used to determine the exact duration of a seizure in the home environment.

## Conclusions

In conclusion, the study did not show any superior benefit of carbogen over pure oxygen in ceasing febrile seizures within 3 minutes. There are possible different reasons that might mask the benefit of carbogen, such as the alkalosis reversal was not confirmed with blood gas analysis, as well as diagnosing a febrile seizure and knowing when the seizure has stopped can be difficult for the parents. Objective detection of the end of a seizure and accurate measurement of seizure

duration are essential for future studies of febrile seizure treatment in the home environment. Wearable devices with a memory function may help to accurately record these essential data.

## Supporting information

**S1 File. Study protocol as PDF file (English translation from the German original).**
(DOCX)

**S2 File. Parents' instruction manual for the use of the study medication.**
(PDF)

**S3 File. CONSORT checklist for the CARDIF trial.**
(DOCX)

## Acknowledgments

We would like to thank the children and their families who participated in this study. We are indebted to the MTI Industrie-Gase AG (Neu-Ulm, Germany) for excellent logistic support, cooperation, development, and the manufacture of the investigational medicinal product. MTI Industrie-Gase AG had no access to the data or the right to decide to publish the results. We sincerely thank the data curators René Bernard and Christina Habermehl from the Excellence Cluster NeuroCure for preparing and anonymizing the original data for public sharing.

## Author contributions

**Conceptualization:** Stephanie Müller-Ohlraun, Dietmar Schmitz, Markus Schuelke.

**Data curation:** Claudia Weiß, Markus Schuelke.

**Formal analysis:** Peter Martus, Markus Schuelke.

**Funding acquisition:** Dietmar Schmitz, Markus Schuelke.

**Investigation:** Claudia Weiß, Alice Hucko.

**Methodology:** Stephanie Müller-Ohlraun, Peter Martus, Dietmar Schmitz, Markus Schuelke.

**Project administration:** Stephanie Müller-Ohlraun, Stefanie Märschenz, Markus Schuelke.

**Resources:** Dietmar Schmitz, Markus Schuelke.

**Software:** Markus Schuelke.

**Supervision:** Dietmar Schmitz, Markus Schuelke.

**Validation:** Markus Schuelke.

**Visualization:** Markus Schuelke.

**Writing – original draft:** Claudia Weiß, Markus Schuelke.

**Writing – review & editing:** Claudia Weiß, Alice Hucko, Stephanie Müller-Ohlraun, Stefanie Märschenz, Peter Martus, Dietmar Schmitz, Markus Schuelke.

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
