## [Decision Letter · Decision Letter 0]

27 May 2025

Dear Dr. Schuelke,

We look forward to receiving your revised manuscript.

Kind regards,

Mohammed Misbah Ul Haq, Pharm-D

Academic Editor

PLOS ONE

Journal Requirements:

2.  We note that the original protocol file you uploaded contains a confidentiality notice indicating that the protocol may not be shared publicly or be published. Please note, however, that the PLOS Editorial Policy requires that the original protocol be published alongside your manuscript in the event of acceptance. Please note that should your paper be accepted, all content including the protocol will be published under the Creative Commons Attribution (CC BY) 4.0 license, which means that it will be freely available online, and any third party is permitted to access, download, copy, distribute, and use these materials in any way, even commercially, with proper attribution.

Therefore, we ask that you please seek permission from the study sponsor or body imposing the restriction on sharing this document to publish this protocol under CC BY 4.0 if your work is accepted. We kindly ask that you upload a formal statement signed by an institutional representative clarifying whether you will be able to comply with this policy. Additionally, please upload a clean copy of the protocol with the confidentiality notice (and any copyrighted institutional logos or signatures) removed.

“The CARDIF trial was funded by the German Research Foundation (DFG, Deutsche Forschungsgemeinschaft) through the German Excellence Strategy (EXC-2049-390688087) via the NeuroCure Consortium at Charité - Universitätsmedizin Berlin to DS and MS and by the Collaborative Research Center SFB 665 TP C4 to DS and MS.”

4. In this instance it seems there may be acceptable restrictions in place that prevent the public sharing of your minimal data. However, in line with our goal of ensuring long-term data availability to all interested researchers, PLOS’ Data Policy states that authors cannot be the sole named individuals responsible for ensuring data access (http://journals.plos.org/plosone/s/data-availability#loc-acceptable-data-sharing-methods).

Reviewers' comments:

Reviewer's Responses to Questions

**Comments to the Author**

1. Is the manuscript technically sound, and do the data support the conclusions?

Reviewer #1: Partly

2. Has the statistical analysis been performed appropriately and rigorously?

Reviewer #1: No

3. Have the authors made all data underlying the findings in their manuscript fully available?

Reviewer #1: Yes

4. Is the manuscript presented in an intelligible fashion and written in standard English?

Reviewer #1: Yes

Reviewer #1: The study hypothesized that “the use of carbogen (5 % CO2 plus 95 % oxygen) in the home environment would be an effective and safe treatment for recurrent febrile seizures”. It used RCT design to test the abovementioned hypothesis.

Here are my comments on the manuscript

1) The primary outcome measure (Seizure-cessation efficacy endpoint) was measured within three minutes of medication application, but the authors did not justify choosing a three-minute window. Why three minutes and not say two or any time greater than 3 minutes? This explanation must be provided in the manuscript. What would have been the implication if the outcome was measured at different time windows??

2) Please include the measurement scales of the secondary outcome measures (binary, count, multinomial, etc.).

3) The section titled statistical analysis is a bit confusing. The content is more of a power analysis (sample size estimation). There should be a section titled Power Analysis to capture the details of sample size estimation, and another section titled Statistical Analysis to highlight the statistical methods used in analysing the data and generating the results in the manuscript.

4) It also does not explicitly state the statistical method(s) that was/were used in quantifying or assessing the relation between the primary outcome measure (Seizure cessation) and the secondary endpoint and the use of carbogen compared to control-100% oxygen. These must be clearly stated

5) In Table 1, the authors indicated, “Age at febrile seizure, median [months]” but rather reported the mean and standard deviation instead of the median and interquartile range. Once again, the t-test does not test the median difference. In addition, normality of the outcome measure required to test mean of a distribution was not assessed and reported

**Do you want your identity to be public for this peer review?** For information about this choice, including consent withdrawal, please see our Privacy Policy

Reviewer #1: No

---

## [Author Response · Author response to Decision Letter 1]

3 Jul 2025

Editor's comments

[1] We note that the original protocol file you uploaded contains a confidentiality notice indicating that the protocol may not be shared publicly or be published. Please note, however, that the PLOS Editorial Policy requires that the original protocol be published alongside your manuscript in the event of acceptance. Please note that should your paper be accepted, all content including the protocol will be published under the Creative Commons Attribution (CC BY) 4.0 license, which means that it will be freely available online, and any third party is permitted to access, download, copy, distribute, and use these materials in any way, even commercially, with proper attribution.

Therefore, we ask that you please seek permission from the study sponsor or body imposing the restriction on sharing this document to publish this protocol under CC BY 4.0 if your work is accepted. We kindly ask that you upload a formal statement signed by an institutional representative clarifying whether you will be able to comply with this policy. Additionally, please upload a clean copy of the protocol with the confidentiality notice (and any copyrighted institutional logos or signatures) removed.

Response: We have now removed the confidentiality notice from the protocol, and being an investigator-initiated trial, we have now added the agreement of the director of our Department (Prof. Dr. Angela Kaindl) to do so. The study protocol can now be published under the CC BY 4.0 license.

[2] Please state what role the funders took in the study. If the funders had no role, please state: "The funders had no role in study design, data collection and analysis, decision to publish, or preparation of the manuscript."

Response: We have added the respective statement into the manuscript.

[3] In this instance it seems there may be acceptable restrictions in place that prevent the public sharing of your minimal data. However, in line with our goal of ensuring long-term data availability to all interested researchers, PLOS’ Data Policy states that authors cannot be the sole named individuals responsible for ensuring data access.

Response: We have now published the original data from the study in an anonymized fashion in accordance with the local regulations and the consent signed by the study participants. The study data set can now be retrieved through https://zenodo.org/records/15728814

[4] Please include captions for your Supporting Information files at the end of your manuscript, and update any in-text citations to match accordingly.

Response: The captions have been added at the end of the manuscript.

Reviewer #1

[1] The primary outcome measure (Seizure-cessation efficacy endpoint) was measured within three minutes of medication application, but the authors did not justify choosing a three-minute window. Why three minutes and not say two or any time greater than 3 minutes? This explanation must be provided in the manuscript. What would have been the implication if the outcome was measured at different time windows??

Response: We have now added the following passage "The ethical requirements imply that no standard treatment must be withheld from the study participants. This would imply the administration of benzodiazepines had the seizure not stopped within the first 3 minutes [25]. We therefore had an observational window of 3 minutes to observe the effect of the study medication, before the standard medication (benzodiazepines) had to be administered."

[2] Please include the measurement scales of the secondary outcome measures (binary, count, multinomial, etc.).

Response: Has been done and added.

[3] The section titled statistical analysis is a bit confusing. The content is more of a power analysis (sample size estimation). There should be a section titled Power Analysis to capture the details of sample size estimation, and another section titled Statistical Analysis to highlight the statistical methods used in analyzing the data and generating the results in the manuscript.

Response: Following the reviewer's suggestion, we have divided the statistical analysis section into two sections (i) Power analysis and (ii) Statistical analysis of endpoints. We have provided the results of the different statistical calculations in Table 1.

[4] It also does not explicitly state the statistical method(s) that was/were used in quantifying or assessing the relation between the primary outcome measure (Seizure cessation) and the secondary endpoint and the use of carbogen compared to control-100% oxygen. These must be clearly stated

Response: We have now added the statistical methods used in the foot line of Table 1.

[5] In Table 1, the authors indicated, “Age at febrile seizure, median [months]” but rather reported the mean and standard deviation instead of the median and interquartile range. Once again, the t-test does not test the median difference. In addition, normality of the outcome measure required to test mean of a distribution was not assessed and reported.

Response: Given the small number of study participants, we have now used a non-parametric test (two-tailed Man-Whitney U-test) and provided the mean and interquartile range for the measurements.

---

## [Decision Letter · Decision Letter 1]

13 Aug 2025

Dear Dr. Schuelke,

Thank you for submitting your manuscript to PLOS ONE. After careful consideration, we feel that it has merit but does not fully meet PLOS ONE’s publication criteria as it currently stands. Therefore, we invite you to submit a revised version of the manuscript that addresses the points raised during the review process.

We look forward to receiving your revised manuscript.

Kind regards,

Dr. Mohammed Misbah Ul Haq, Pharm-D

Academic Editor

PLOS ONE

Journal Requirements:

Reviewers' comments:

Reviewer's Responses to Questions

**Comments to the Author**

Reviewer #1: All comments have been addressed

Reviewer #2: (No Response)

Reviewer #3: (No Response)

2. Is the manuscript technically sound, and do the data support the conclusions?

Reviewer #1: Yes

Reviewer #2: Yes

Reviewer #3: Yes

3. Has the statistical analysis been performed appropriately and rigorously?

Reviewer #1: Yes

Reviewer #2: Yes

Reviewer #3: Yes

4. Have the authors made all data underlying the findings in their manuscript fully available?

Reviewer #1: Yes

Reviewer #2: Yes

Reviewer #3: Yes

5. Is the manuscript presented in an intelligible fashion and written in standard English?

Reviewer #1: Yes

Reviewer #2: Yes

Reviewer #3: Yes

Reviewer #1: The authors have addressed all my previous comments and the content of the revised manuscript has improved significantly

Reviewer #2: The use of carbogen for the interruption of febrile seizures - the randomized controlled CARDIF trial

Summary

The authors did a clinical trial based on animal studies. They targeted one of the most common pediatric diseases, which has significant long-term and short-term complications. The study also aligns with other promising studies being conducted to assess the effectiveness of carbogen in treating absence seizures and status epilepticus. They clearly outlined the rationale for doing the study.

Questions

1. What proportion of the patients in your study had complex febrile seizures, and what percentage had simple febrile seizures?

2. Using parent-reported seizure duration, benzodiazepine use, and any serious adverse events is understandable, but wouldn't it be better to also add home video recording since at times patients can have a seizure attack but the parents may not notice it if they are not around, or if they are sleeping.

3. You need to add to the manuscript why the 5% carbondioxide and 95% oxygen are used. What were your reasons to avoid 7%/93% or other proportions?

4. When the low-pressure can was used, what precautions were taken to avoid rebreathing? The reason that I asked this is that even if the carbogen composition is 5% CO2 and 95% O2, the expiratory air has 4% CO2 and 16% O2 composition, which can affect the results of your study.

5. What the caregivers were advised, they might forget, so giving written advice will be beneficial. Was written instruction given for the caregivers?

6. How can we reliably say that the abnormal body movement the child has at home is a febrile seizure? What if the child is having shivering, and the parents think it is a seizure? What if the child is not having a fever at the time of the seizure that occurred at home? What if the child is having a central nervous system infection?

7. What if the child has seizures and the family didn't report? How do you make sure that the families report and apply the carbogen or 100% oxygen whenever their child has a febrile seizure?

8. How is the safety profile of the equipment? Is there a possibility that the equipment is malfunctioning and stopping air flow, which can potentially lead to rebreathing or suffocation?

9. In line 106, you stated that the study was done between 2012 and 2015. What was the reason for not submitting the result and conclusion earlier?

10. What was the reason for making the face mask loose-fitting?

Comments

The conclusion is not well suited for the results and the discussion of the study. A better conclusion would be: The study didn't show any superior benefit of carbogen over pure oxygen in ceasing febrile seizures within 3 minutes. There are possible different reasons that might mask the benefit of carbogen, such as the alkalosis reversal was not confirmed with blood gas analysis, diagnosing febrile seizure can be difficult for parents, and knowing when the seizure has stopped is difficult for the parents.

Reviewer #3: Review to paper “The use of carbogen for interruption of febrile seizures - the randomized controlled CARDIF trial”

As a general comment, I want to say that this work meets all the criteria I was asked to evaluate. It is original, unpublished, and carried out with a high level of technical rigor, with the methods, analyses, and conclusions clearly described and supported by the data. The paper is well written, ethically sound, and follows the expected reporting standards. More importantly, I found it both relevant and informative, and I truly appreciate how creative and thorough the team has been in tackling the complex logistics, preparation, and follow-up for such a challenging study. Even as an interim analysis, the findings have real potential to move the field forward. However, I do have some minor and a few major comments that I believe could be of value to the authors.

Minor comments

1. In the abstract, on page 3, lines 43–44, it is not clear whether you are referring to the number of children or the number of febrile seizures: “The febrile seizure was terminated in 5/15 children on carbogen and in 8/11 children on oxygen.” I understood that the study medication was used in n = 30 febrile seizures (15 episodes in children within the verum arm, 11 episodes in children within the placebo arm, and 4 episodes in children within the verum open arm), in a total of 20 children. Check whether this could be clarified.

2. On page 6, line 113, the sentence could be improved because it contains two actions (“have been published” and “are added”) that are not connected by a linking word such as “and” or “which.”

3. In Table 1, page 9, line 233, the number of patients in the verum arm indicates that 32 [69.6%] have one FS in history and 15 [32.6%] have more than one FS in history, which does not add up to 46. Verify if this is correct, since cases per gender indeed add up to 46.

4. It would be stronger and clearer if, in the Methods section corresponding to “Randomisation and allocation,” the final number of patients randomised and followed, as well as those with recurrent febrile seizures, were stated. Additionally, it would be beneficial to refer to the CONSORT flowchart provided.

5. The section on power analysis, in my opinion, should not be part of the statistical analysis since it is mostly related to sample size determination, either as part of the explanation on how a desired sample size was decided or to contextualize the limitations faced during the interim phase of the clinical trial. It is not an actual analysis of the data obtained. An easy solution would be to put it as an individual subsection within the Methods.

6. What is the reason for carrying out a conventional chi-squared test instead of a Fisher’s exact test? You have a small sample size; usually, Fisher’s exact test works better under such conditions (small sample size or when any expected cell count is <5). However, looking at your results, it would not make much of a difference. However, since you actually followed that kind of reasoning when you applied the Mann–Whitney U test, you should consider this.

7. When comparing the patients in the crossover part of the study (6 patients in each group), the reasoning behind using a statistical test (regardless of whether it is a chi-squared test or a Mann–Whitney U test) is even more confusing. In my opinion, it is not worth relying on a formal null-hypothesis test with n = 6 vs. 6. You can run one, but the result will be practically uninterpretable and may mislead readers. It would be better to treat the data as descriptive/exploratory and focus on effect sizes, exact confidence intervals, and raw data. Formal testing is not useful here since the expected power is extremely low for any but enormous effects, so a nonsignificant p-value mostly reflects small n, not “no effect.” You already followed this logic in Table 2.

8. You include in your paper the CONSORT flowchart for participation; however, in no other place in the paper do you mention adherence to the CONSORT guidelines. If you adhered to any reporting guidelines, and also to guidelines for designing the trial (or good practice guidelines), you should mention it.

Major comments

1. In relation to how the hypothesis is declared

I have some issues with the way the hypothesis is stated (page 6, lines 117–119), especially from a reader’s standpoint before reading the rest of the Methods section.

You stated: “…Six liters of carbogen gas administered over a period of 3 minutes through a low-pressure can with an attached loose-fitting face mask is safe and more effective than placebo (100% oxygen) in interrupting recurrences of febrile seizures…”

I assumed, based on what you explained throughout the paper, that the intention is to communicate that you want to assess whether the intervention stops the seizure during the 3-minute window while receiving the intervention.

However, in my opinion, readers could understand that the intention is to communicate that the intervention’s goal is to prevent another seizure episode after receiving the intervention (3 minutes’ exposure to carbogen). The wording “interrupting recurrence” may imply this. If you say that the outcome of interest is “interrupting recurrences of febrile seizures” to a clinician, it would usually be understood as preventing another seizure from happening after the episode under treatment.

For the reader, this could mean both stopping the ongoing seizure and giving a treatment that reduces the likelihood of subsequent seizures. I am sure your intention is to communicate as the outcome “stopping of an ongoing seizure” rather than preventing a new one; in that case, you are measuring termination.

I think readers would benefit from a clearer statement of the hypothesis, similar to how the aim is stated: to determine whether 6 liters of carbogen gas (5% carbon dioxide + 95% oxygen) delivered from a low-pressure can would suppress acute febrile seizures (page X, lines 102–104).

2. In relation to the power analysis

If, in the most conservative scenario, you expected a power of 86% for a difference of 50% (75% vs. 25%), then when, in the preliminary phase, you obtained 15 episodes in one arm and 11 in the other, that would indeed impact power. This should be addressed either in the power analysis section (my recommendation) or somewhere in the Results (as a limitation within the interim results).

3. In relation to reason for not observing the expected effect

In the Discussion, you are right to discuss the reasons why you did not observe a similar effect as previously reported by small hospital-based studies or animal studies, and I agree with you that difficulties in the administration of carbogen are a major determinant of not achieving proper exposure under such a critical situation. However, you should briefly discuss the possibility that carbogen (with 5% CO₂) is not actually effective in children. Is there any reason you may think this could be possible?

Steven N. Cuadra

Steven.cuadra@gmail.com

**Do you want your identity to be public for this peer review?** For information about this choice, including consent withdrawal, please see our Privacy Policy

Reviewer #1: No

Reviewer #2: No

Reviewer #3: **Yes: ** Steven Napoleón Cuadra

---

## [Author Response · Author response to Decision Letter 2]

18 Sep 2025

Reviewer #1

The authors have addressed all my previous comments and the content of the revised manuscript has improved significantly

Response: We thank the reviewer for the effort to read and suggest improvements to our manuscript.

Reviewer #2

[1] What proportion of the patients in your study had complex febrile seizures, and what percentage had simple febrile seizures?

Response: Of the total number of seizures, n = 32 were simple febrile seizures and n = 6 were complex febrile seizures. Of those treated with verum, n = 12 were simple and n = 3 were complex febrile seizures; all n = 11 febrile seizures treated with placebo were simple febrile seizures. The n = 3 complex febrile seizures in the verum arm were not interrupted by verum. We have now entered this information into the CONSORT flow chart.

Response: The difference in complex versus simple febrile seizures could indeed partly explain the negative outcome of the study. We thank the reviewer for this important comment. We therefore added this finding to the result section (“In the verum arm n = 12 recurrent febrile seizures were simple and n = 3 were complex, whereas all recurrent febrile seizures in the placebo arm were simple.”) and a sentence to the discussion section (“Furthermore, the difference in complex versus simple febrile seizures in both study arms may have influenced the outcome as complex febrile seizures only occurred in the verum arm.”).

[2] Using parent-reported seizure duration, benzodiazepine use, and any serious adverse events is understandable, but wouldn't it be better to also add home video recording since at times patients can have a seizure attack but the parents may not notice it if they are not around, or if they are sleeping.

Response: Indeed it would have been ideal to have some video footage. However, first we have to consider that the study was conducted ten years ago when mobile phones with video capability had not yet been ubiquitous during everyday life, and secondly during a seizure, parents tended to be agitated and it would have been asking too much from them to apply the carbogen bottle and at the same time video record their child. Equally a continuous video monitoring of the children during sleep would have been technically not feasible.

[3] You need to add to the manuscript why the 5% carbon dioxide and 95% oxygen are used. What were your reasons to avoid 7%/93% or other proportions?

Response: In the experiments leading to our medicinal product (the 6 liter carbogen bottle with attached mask) we have performed time and concentration serial experiments to achieve an inspiratory pCO2 partial pressure of 5 mmHg. This was the partial pressure that was efficient in the animal experiments and in the single incidences where carbogen had been successfully administered in single cases under EEG-monitoring.

[4] When the low-pressure can was used, what precautions were taken to avoid rebreathing? The reason that I asked this is that even if the carbogen composition is 5% CO2 and 95% O2, the expiratory air has 4% CO2 and 16% O2 composition, which can affect the results of your study.

Response: The design of the mask (loose fitting, small covered volume) and the continuous stream of fresh carbogen gas from the bottle prevented any rebreathing effect, which we had tested in our preliminary experiments in the run up to the study.

[5] What the caregivers were advised, they might forget, so giving written advice will be beneficial. Was written instruction given for the caregivers?

Response: We had provided a written and illustrated instruction to the parents how to use the bottle. This had also been translated in various languages to make sure that the parents understood the handling of the bottle.

[6] How can we reliably say that the abnormal body movement the child has at home is a febrile seizure? What if the child is having shivering, and the parents think it is a seizure? What if the child is not having a fever at the time of the seizure that occurred at home? What if the child is having a central nervous system infection?

Response: As mentioned in the paper, there was of course always a certain margin of uncertainty because we had to rely on patient reported outcomes. However, this uncertainty cannot be completely removed as it was in the nature of the study. As the parents had already experienced a seizure in their children before they were recruited to the study We thus assumed that the parents would be knowledgeable, how a seizure would look like in their child. To make sure that it was a febrile seizure, parents were instructed to measure the body temperature and report the fact that a seizure had occurred and the body temperature at the time of the seizure to the study team.

[7] What if the child has seizures and the family didn't report? How do you make sure that the families report and apply the carbogen or 100% oxygen whenever their child has a febrile seizure?

Response: As mentioned above, we were dependent on the parents to report any incidence where the child had a presumed seizure or when the study medication was used. Indeed, there have been incidents where seizures occurred and the parents did not apply the study medication (e.g. because they were shopping and had forgotten to take the bottle with them). These incidents were recorded in the patient case files, reported in the CONSORT flow sheet, but not included into the final efficacy analysis.

[8] How is the safety profile of the equipment? Is there a possibility that the equipment is malfunctioning and stopping air flow, which can potentially lead to rebreathing or suffocation?

Response: The bottles were tested and certified by the BfArM, which is the public entity in Germany responsible for the admission of medicinal products. As mentioned above, the mask was loose-fitting and the covered volume above the nose was small so that no rebreathing could occur. We have tested the bottle/mask combination in place without any gasflow and did not see a decline in oxygen saturation (measured by pulse oximetry) or an increase of transcutaneous pCO2 values.

[9] In line 106, you stated that the study was done between 2012 and 2015. What was the reason for not submitting the result and conclusion earlier?

Response: It was a negative finding and at the first place we had tried several pediatric journals, only to be rejected (despite the fact that those journals had also publicly stated to publish negative results). Then, honestly, we thought it would not be worth the while to pursue publishing our negative results. However, later we came to the conclusion that publication would be necessary, because the study had been registered in ClinicaTrials.gov and colleagues kept asking by email what had come out of the study.

[10] What was the reason for making the face mask loose-fitting?

Response: The mask had to be loose-fitting to prevent potential rebreathing and suffocation. The flow allowed to achieve 5 mmHg inspiratory pCO2 pressure over 3 minutes.

[11] The conclusion is not well suited for the results and the discussion of the study. A better conclusion would be: The study didn't show any superior benefit of carbogen over pure oxygen in ceasing febrile seizures within 3 minutes. There are possible different reasons that might mask the benefit of carbogen, such as the alkalosis reversal was not confirmed with blood gas analysis, diagnosing febrile seizure can be difficult for parents, and knowing when the seizure has stopped is difficult for the parents.

Response: We thank the reviewer for the suggestion and have amended our conclusion accordingly.

Reviewer #3:

Review to paper “The use of carbogen for interruption of febrile seizures - the randomized controlled CARDIF trial”

Minor comments

[1] In the abstract, on page 3, lines 43–44, it is not clear whether you are referring to the number of children or the number of febrile seizures: “The febrile seizure was terminated in 5/15 children on carbogen and in 8/11 children on oxygen.” I understood that the study medication was used in n = 30 febrile seizures (15 episodes in children within the verum arm, 11 episodes in children within the placebo arm, and 4 episodes in children within the verum open arm), in a total of 20 children. Check whether this could be clarified.

Response: We rephrased this sentence as follows: “The febrile seizure was terminated in 5/15 episodes on carbogen and in 8/11 episodes on oxygen (Fischers' exact test; p = 0.11).”

[2] On page 6, line 113, the sentence could be improved because it contains two actions (“have been published” and “are added”) that are not connected by a linking word such as “and” or “which.”

Response: “and” has been added.

[3] In Table 1, page 9, line 233, the number of patients in the verum arm indicates that 32 [69.6%] have one FS in history and 15 [32.6%] have more than one FS in history, which does not add up to 46. Verify if this is correct, since cases per gender indeed add up to 46.

Response: Thank you for pointing out this mistake. It should be 14 [30.4%] patients with more than one FS in history. This has now been corrected in Table 1.

[4] It would be stronger and clearer if, in the Methods section corresponding to “Randomisation and allocation,” the final number of patients randomised and followed, as well as those with recurrent febrile seizures, were stated. Additionally, it would be beneficial to refer to the CONSORT flowchart provided.

Response: The reference to the CONSORT flowchart has now been added. The chart has been amended and now provides the above asked information. Additionally, the number of patients with simple and complex febrile seizures in the treatment and placebo arms are now mentioned separately and discussed (see answer to reviewer #2).

[5] The section on power analysis, in my opinion, should not be part of the statistical analysis since it is mostly related to sample size determination, either as part of the explanation on how a desired sample size was decided or to contextualize the limitations faced during the interim phase of the clinical trial. It is not an actual analysis of the data obtained. An easy solution would be to put it as an individual subsection within the Methods.

Response: We agree and have now moved the "Power analysis" section into the Methods section / Study design.

[6] What is the reason for carrying out a conventional chi-squared test instead of a Fisher’s exact test? You have a small sample size; usually, Fisher’s exact test works better under such conditions (small sample size or when any expected cell count is <5). However, looking at your results, it would not make much of a difference. However, since you actually followed that kind of reasoning when you applied the Mann–Whitney U test, you should consider this.

Response: We agree with the reviewer and have now used the Fischer's exact test for all respective calculations.

[7] When comparing the patients in the crossover part of the study (6 patients in each group), the reasoning behind using a statistical test (regardless of whether it is a chi-squared test or a Mann–Whitney U test) is even more confusing. In my opinion, it is not worth relying on a formal null-hypothesis test with n = 6 vs. 6. You can run one, but the result will be practically uninterpretable and may mislead readers. It would be better to treat the data as descriptive/exploratory and focus on effect sizes, exact confidence intervals, and raw data. Formal testing is not useful here since the expected power is extremely low for any but enormous effects, so a nonsignificant p-value mostly reflects small n, not “no effect.” You already followed this logic in Table 2.

Response: We agree with the reviewer and have now only described the outcomes of the cross-over arm by mentioning the median and interquartile range.

[8] You include in your paper the CONSORT flowchart for participation; however, in no other place in the paper do you mention adherence to the CONSORT guidelines. If you adhered to any reporting guidelines, and also to guidelines for designing the trial (or good practice guidelines), you should mention it.

Response: The following sentence has been added to the methods section: The study was conducted according to the guidelines of Consolidated Standards of Reporting Trials (CONSORT) and good clinical practice. The filled in CONSORT questionnaire is now attached to the supplementary materials.

Major comments

[9] In relation to how the hypothesis is declared: I have some issues with the way the hypothesis is stated (page 6, lines 117–119), especially from a reader’s standpoint before reading the rest of the Methods section. You stated: “…Six liters of carbogen gas administered over a period of 3 minutes through a low-pressure can with an attached loose-fitting face mask is safe and more effective than placebo (100% oxygen) in interrupting recurrences of febrile seizures…” I assumed, based on what you explained throughout the paper, that the intention is to communicate that you want to assess whether the intervention stops the seizure during the 3-minute window while receiving the intervention. However, in my opinion, readers could understand that the intention is to communicate that the intervention’s goal is to prevent another seizure episode after receiving the intervention (3 minutes’ exposure to carbogen). The wording “interrupting recurrence” may imply this. If you say that the outcome of interest is “interrupting recurrences of febrile seizures” to a clinician, it would usually be understood as preventing another seizure from happening after the episode under treatment. For the reader, this could mean both stopping the ongoing seizure and giving a treatment that reduces the likelihood of subsequent seizures. I am sure your intention is to communicate as the outcome “stopping of an ongoing seizure” rather than preventing a new one; in that case, you are measuring termination. I think readers would benefit from a clearer statement of the hypothesis, similar to how the aim is stated: to determine whether 6 liters of carbogen gas (5% carbon dioxide + 95% oxygen) delivered from a low-pressure can would suppress acute febrile seizures (page X, lines 102–104).

Response: We agree that the wording might be easily misunderstood. As we only recruited children with at least one previous febrile seizure, any following seizure would be considered a "recurrence". The aim of the study was clearly to interrupt the ongoing seizure within the 3 minutes of carbogen application. We have changed the text to make this clear.

[10] In relation to the power analysis: If, in the most conservative scenario, you expected a power of 86% for a difference of 50% (75% vs. 25%), then when, in the preliminary phase, you obtained 15 episodes in one arm and 11 in the other, that would indeed impact power. This should be addressed either in the power analysis section (my recommendation) or somewhere in the Results (as a limitation within the interim results).

Response: We have added a sentence to the power analysis section that an unequal distribution of seizure events between treatment arms, which might occur in studies with a low number of patients / treatment events might impact the power analysis and increase the α (type I) error.

[11] In relation to reason for not observing the expected effect: In the Discussion, you are right to discuss the reasons why you did not observe a similar effect as previously reported by small hospital-based studies or animal studies, and I agree with you that difficulties in the administration of carbogen are a major determinant of not achieving proper exposure under such a critical situation. However, you should briefly discuss the possibility that carbogen (with 5% CO₂) is not actually effective in children. Is there any reason you may think this could be possible?

Response: Children with febrile seizures clearly have a higher frequency of increased blood pH values (alkalosis) due to hyperventilation as shown by Schuchmann et al. 2011 (PMID 21910730, Figure 1A). Using added

---

## [Editor Report · Decision Letter 2]

25 Nov 2025

The use of carbogen for interruption of febrile seizures - the randomized controlled CARDIF trial

PONE-D-25-19281R2

Dear Dr. Schuelke,

We’re pleased to inform you that your manuscript has been judged scientifically suitable for publication and will be formally accepted for publication once it meets all outstanding technical requirements.

Kind regards,

Dr. Mohammed Misbah Ul Haq, Pharm-D

Academic Editor

PLOS ONE
---

## [Editor Report · Acceptance letter]

PONE-D-25-19281R2

PLOS One

Dear Dr. Schuelke,

I'm pleased to inform you that your manuscript has been deemed suitable for publication in PLOS One. Congratulations! Your manuscript is now being handed over to our production team.

Kind regards,

on behalf of

Dr. Mohammed Misbah Ul Haq

Academic Editor

PLOS One